# Bicuspid Aortic Valve in Children and Adolescents: A Comprehensive Review

**DOI:** 10.3390/diagnostics12071751

**Published:** 2022-07-20

**Authors:** Gaia Spaziani, Francesca Girolami, Luigi Arcieri, Giovanni Battista Calabri, Giulio Porcedda, Chiara Di Filippo, Francesca Chiara Surace, Marco Pozzi, Silvia Favilli

**Affiliations:** 1Pediatric and Transition Cardiology, Meyer Children’s Hospital, Viale Pieraccini 24, 50139 Florence, Italy; francesca.girolami@meyer.it (F.G.); giovanni.calabri@meyer.it (G.B.C.); giulio.porcedda@meyer.it (G.P.); c.difilippo91@gmail.com (C.D.F.); silvia.favilli@meyer.it (S.F.); 2Pediatric Cardiology and Cardiac Surgery, Ospedali Riuniti, Via Conca 71, 60126 Ancona, Italy; luigi.arcieri@ospedaliriuniti.marche.it (L.A.); francescachiara.surace@ospedaliriuniti.marche.it (F.C.S.); marco.pozzi@ospedaliriuniti.marche.it (M.P.)

**Keywords:** bicuspid aortic valve, children, diagnostic imaging, echocardiography, natural history

## Abstract

Bicuspid aortic valve (BAV) is the most common congenital heart defect. Prevalence of isolated BAV in the general pediatric population is about 0.8%, but it has been reported to be as high as 85% in patients with aortic coarctation. A genetic basis has been recognized, with great heterogeneity. Standard BAV terminology, recently proposed on the basis of morpho-functional assessment by transthoracic echocardiography, may be applied also to the pediatric population. Apart from neonatal stenotic BAV, progression of valve dysfunction and/or of the associated aortic dilation seems to be slow during pediatric age and complications are reported to be much rarer in comparison with adults. When required, because of severe BAV dysfunction, surgery is most often the therapeutic choice; however, the ideal initial approach to treat severe aortic stenosis in children or adolescents is not completely defined yet, and a percutaneous approach may be considered in selected cases as a palliative option in order to postpone surgery. A comprehensive and tailored evaluation is needed to define the right intervals for cardiologic evaluation, indications for sport activity and the right timing for intervention.

## 1. Introduction

Bicuspid aortic valve (BAV) is the most common congenital malformation; it may occur both as an isolated lesion or in association with congenital heart disease (CHD), mainly aortic coarctation and/or other left ventricular (LV) obstructive lesions. Incidence of isolated BAV in the general population is about 1%, while a prevalence of about 0.5–0.8% has been reported in healthy school children and young adults [1,2,3]. Prevalence in patients with aortic coarctation is is as high as 50–85%; conversely, BAV prevalence in patients with other CHD, such as septal defects or right heart obstructions, is similar to that reported in the general population [4]. There is a strong association with Turner syndrome, where BAV has been described in 15–30% of patients [5]. 

## 2. BAV: Genetic Insights

Genetic heterogeneity, with a complex genetic aetiology, has been described by recent studies [6,7]. Several studies have demonstrated the high heritability of BAV (as well as of other left-sided CHD), with a prevalence in first-degree relatives of BAV patients 10-fold higher than in the general population [8,9]. However, there is not enough evidence to support a family screening involving paediatric subjects in the absence of clinical signs suggestive of valve disease [10]. 

BAV is a complex disorder inherited in an autosomal dominant pattern with low penetrance, variable expressivity and male predominance (3:1). We can distinguish isolated forms from genetic syndromes with BAV. These last manifestations are primarily Turner (where BAV is present in from 15% to 30% of Turner women), Loeys–Dietz, Marfan and velocardiofacial syndromes; in other genetic disorders such as Down, Kabuki, Andersen or Alagille syndromes, BAV is a very rare cardiac feature. In non-syndromic BAV, other CHD are usually present, and even if the genetic architecture involves many different genes, the heritability is about 90% [11,12]. There are some genes for which an association with non-syndromic BAV has been demonstrated: NOTCH1, SMAD6, GATA4, GATA5, GATA6, ROBO4, MAT2A, ADAMTS19, TBX20 and NKX2-5; each of these genes explains only a small percentage of the overall non-syndromic BAV prevalence and involves different molecular pathways [6] where SNVs (single-nucleotide variants) and CNVs (copy number variants) can be detected with routine molecular techniques such as NGS (next-generation sequencing) or SNPs (single-nucleotide polymorphisms) array. For genetic testing, the careful analysis of pedigree in large families during genetic counselling is an essential step to determine the inheritance pattern. At present, the indication for genetic testing is limited to selected cases: in the case of syndromic BAV or where there is the suspicion of a single-gene aetiology or in high–risk patients, for example, with other CHD or history of sudden death. In unaffected minors there is no indication to perform cascade predictive tests [13]. 

## 3. Morphology and Classification

Recently, a standard terminology mainly based on aortic valve imaging with transthoracic echocardiography (TTE)was proposed by Michelena and colleagues [14]. Three phenotypes were identified: (1) fused BAV—right-to-left (R-L) cusp fusion, right-non coronary (R-N) cusp fusion, left-non coronary (L-N) cusp fusion and indeterminate phenotype; (2) two-sinus BAV (latero-lateral and antero-posterior phenotypes); and (3) partial fused BAV. This morphologic classification may be easily applied to the paediatric population.

In a large multicentre retrospective study (MIBAVA Consortium) involving more than 2000 children with BAV (mean age 10.2 years), the most common morphology was R-L fusion (65.7%), followed by R-N fusion (32.9%) [15]. BAV morphology has been related to the progression of valve dysfunction or aortic dilation and to the association with additional CHD [4]. Paediatric patients with left heart obstructive lesions were more commonly reported to present a right to left (R-L) cusp fusion [16]. In the MIBAVA Consortium study, R-L fusion was associated with aortic coarctation, while R-N fusion was associated with valve dysfunction (stenosis and/or regurgitation).

Dilation at the level of the proximal ascending aorta is largely prevalent in children and adolescents with BAV and was found in approximately 50% of patients in paediatric BAV populations [15,16]. Both haemodynamic and genetic factors have been advocated to explain the association with aortic disease. In the MIBAVA Consortium study, R-N fusion was independently associated with dilation of the ascending aorta, suggesting a predominant role of genetic predisposition in these patients. However, haemodynamic factors seem to play an important role, too, in other cases, as a significant aortic valve regurgitation has been reported to be associated with larger diameters at the Valsalva sinus level [15]. Ascending aorta dilation increases with age [15,16,17]. Due to the peculiar aortic morphology, dedicated aortic nomograms for children and adolescents with BAV could be useful for monitoring this progression [15]. 

Several variations in coronary artery (CA) anatomy, mainly separate ostia of the left ascending and circumflex CA, have been described in patients with BAV. In children who present BAV associated with complex left heart CHD, a high take-off of CA (described in nearly 25% of cases and potentially related to harmful complications) should be excluded [18].

## 4. Clinical Presentation in Different Paediatric Ages and Natural History

BAV is the main cause of valvular stenosis in children and adolescents. Critical neonatal aortic stenosis (AS), which presents with low output heart failure and requires a prompt release of LV obstruction, has a peculiar outcome and is often associated with other LV obstructive lesions. Therefore, stenotic BAVs with clinical presentation in the first year of age are usually not considered/excluded in studies concerning the natural history of BAV in paediatric populations [16,19]. In the paediatric ages that follow, children with BAV are often asymptomatic and presentation is more commonly with systolic and/or diastolic murmur or a protosystolic sound.

Other than for the first year of age, more-than-moderate AS is reported in a minority of paediatric patients. A mild to moderate aortic regurgitation (AR) is found in the majority of children and adolescents with BAV, but progression to severe AR is uncommon during paediatric age [16].

Children with isolated BAV and no or mild valve dysfunction at diagnosis usually have little disease progression before adulthood [20]. In a recent single-centre, paediatric study, only one patient required surgery because a progressive aortic regurgitation and only 3% developed more than mild stenosis or regurgitation [20]. 

Progression of aortic dilation seems to be relatively slow during paediatric age. In a recent series [20], an average increase of 1.00 mm/year in aortic root and ascending aorta was described; such a low progression was confirmed also in infancy and adolescence, which are both characterized by high somatic growth. Both entity and progression of aortic dilation are lower in BAV associated with aortic coarctation than in isolated BAV; AR was associated with greater diameters of proximal ascending aorta [17]. 

In a large paediatric population, including subjects with BAV aged from 0 to 20 years, the strongest predictors of progressive dilation of the proximal ascending aorta were severe aortic stenosis and moderate or severe aortic regurgitation. The risk of developing a significant aortic dilation in early adulthood was 9-fold higher in patients with both more-than-mild stenosis and regurgitation, suggesting a strong influence of hemodynamic factors [21]. On the contrary, a very slow aortic dilatation rate was reported in children with a normally functioning BAV, independently from BAV leaflet fusion type. However, a significant dilation was detected in some young patients with normally functioning BAV, which might be related to degenerating processes involving the aortic wall, even in the absence of haemodynamic stress [21].

In comparison with adults, children with BAV have fewer complications during follow-up, including interventions for valve dysfunction and aortic dilation [16,19,20]. No cases of aortic dissection have been reported in the literature. Mahle et al., in a large series of paediatric patients with BAV, reported an event rate of 0.004/patient-years, with only one case of infective endocarditis and no cases of dissection [19]. A very low rate of complications was found also in a paediatric population with BAV followed at Meyer Children’s Hospital, in which no cases of dissection or endocarditis were observed [16].

Despite the rarity of complications, children and adolescents with isolated uncomplicated BAV are often followed at short-term intervals, probably because of the concern about the possible evolution of valve dysfunction.

## 5. Diagnostic Imaging

TTE represents the first-line method to establish the diagnosis of BAV and to assess the presence and severity of valve dysfunction and/or associated aortopathy. Diagnosis is based on the recognition of a ‘fish-mouth’ appearance of the orifice in systole (Figure 1). The different morphologic patterns are defined on the basis of the new classification proposed for adults with BAV [14]. The presence, extension and number of raphe should be described. Calcifications are rare but may occasionally occur also in adolescents with BAV and should raise the suspicion of a genetic predisposing mutation [22].

The assessment of an aortic leaflet prolapse, which has been associated with valve regurgitation and may have implications on a possible valve repair surgery, can be suspected by TTE but usually requires confirmation by transoesophageal echocardiography also in paediatric ages [23].

A differential diagnosis should be made between BAV and unicuspid aortic valve (UAV). UAV is a rare condition which accounts for 4–5% of cases undergoing valve surgery for AS and is a frequent cause of intervention in younger patients (<25 years) [24,25]. In paediatric patients, UAV may often be diagnosed by TTE looking at the parasternal short axis view of the aortic valve due to the usually good-quality images and lack of valve calcification (Figure 2); however, three-dimensional echocardiography has potential advantages over conventional TTE in confirming the diagnosis as well as in the monitoring of interventional procedures [24].

The ascending aorta is evaluated in parasternal long axis view at four different levels (aortic annulus, sinuses of Valsalva, sino-tubular junction and proximal ascending aorta). Due to the usually favorable acoustic window in children, TTE is suitable for assessing aortic diameters using the leading edge–,leading edge convention in end diastole for all measurements, but the aortic annulus is usually measured with the inner edge–inner edge convention in systole. In children, values must be indexed for age and surface area by using nomograms developed for paediatric populations [26].

Recently, a new tool based on a machine learning algorithm (Q score) has been proposed for assessing normalcy of the thoracic aorta [27]. The Q score was developed on a cohort of healthy subjects comprising a subgroup of children aged > 5 years. The new score potentially provides a more comprehensive evaluation of aortic geometry, avoiding over-diagnosis of ‘aortopathy’, which may have negative consequences in terms of quality of life, especially in adolescents. Sensitivity and specificity assessment require its use in larger paediatric populations with BAV, with a long-term follow-up.

Aortic coarctation, which is associated with 2% of cases of BAV, should be always ruled out by assessing aortic arch in the suprasternal view and the isthmus in the parasagittal high left parasternal view and by evaluating pulsatility and flow of the abdominal aorta.

Cardiac magnetic resonance (CMR) is usually performed in older children and adolescents with dilation of aortic root or, more commonly, of ascending aorta > 4 DS (Figure 3). If the measurements are comparable with the TTE values, then follow-up can usually be performed using echocardiography. 

After surgery or in patients with valve dysfunction, CMR may be useful for a more comprehensive assessment of ascending aorta, LV dimension and function and quantification of AR. Polte and colleagues reported a characterization of severe AR by a regurgitant volume (RVol) > 40 mL (or >20 mL/m^2^) and a regurgitation fraction (RF) >30% (direct method by aortic flow) and RVol > 62 mL (or >31 mL/m^2^) and RF > 36% (indirect method by LV and pulmonary stroke volume) [28]. Although CMR studies regarding AR severity and LV volume are still scarce, Gao and colleagues proposed cut-off values for LV end-diastolic volume of 251 mL (or 127 mL/m^2^) for severe AR, defined by RF > 33% [29]. Moreover, the holodiastolic retrograde flow in the descending aorta by CMR is significantly associated with outcome in AR patients [30].

Decreased LV global longitudinal strain (GLS) assessed by CMR was reported by Stefek et al. [31] in a small cohort of post-surgical BAV patients with normal ejection fraction (EF). Altered aortic haemodynamics (high peak systolic velocity and peak wall shear stress in ascending aorta, abnormal flow patterns) were related to an adverse LV remodelling. 

GLS is becoming a useful echocardiographic tool to assess initial LV dysfunction when EF is still within the normal range, leading to a growing interest in its potential role in guiding the management of valvular heart diseases. In asymptomatic patients with at least moderate AR and preserved EF, reduced GLS has been related with a higher long-term mortality in those patients who did not undergo aortic surgery [32]. Similarly, impaired GLS has been described in asymptomatic patients with severe AS although a preserved EF: the reduced GLS was associated with a higher risk of developing symptoms and with the need of aortic valve intervention [33]. A cut-off of 14.7% has independently been associated with the increased risk of death, underlining a potential prognostic role of GLS in asymptomatic significant AS [34]. In paediatric patients with congenital AS treated with balloon aortic valvuloplasty, post-procedural GLS was lower than in healthy children, with more impaired values in those with LV eccentric hypertrophy or with higher residual aortic gradients [35]. Carlos et al. reported lower GLS values in BAV patients [36]. Impaired GLS has been observed more frequently in cases of valve dysfunction, with a progressive worsening according to the severity of the valvulopathy, and it has been correlated, even in BAV patients, with the risk of undergoing aortic valve replacement [37]. The potential prognostic impact of GLS, unmasking a subclinical systolic dysfunction, may help in risk stratifying in order to guide the optimal timing of aortic valve replacement.

## 6. Clinical Issues

### 6.1. Antibiotic Prophylaxis

Currently, antibiotic prophylaxis to prevent infective endocarditis (IE) is not recommended for isolated BAV, which is not considered a condition at high risk. However, there is not a general agreement about this policy as IE incidence in BAV is reported to be ~30-fold higher than in the general population, frequently from suspected odonatological origin [38]. Many pediatric cardiologists in Europe and in the United States continue to prescribe antibiotic prophylaxis even if doing so is not recommended by the guidelines [39].

Piercing and tattooing enjoy large popularity among adolescents. Although published cases with infective complications are sporadic, young patients with BAV should be strongly advised, especially against piercing and tattoos, with regard to the potential risk of IE. In any case, the correct advice concerning antibiotic prophylaxis and prompt care in case local infection occurs should be provided.

### 6.2. Sport Activity

Following guidelines, the diagnosis of normally functioning BAV in children and adolescents with normal aortic diameters does not affect the eligibility for competitive sport [40]. However, in subjects with aortic dilation, there is concern about a possible negative role of intense physical exercise and/or competitive sport on the natural course of aortopathy, so increasing the potential risk of rupture or dissection. Strict adherence to guidelines could produce inappropriate restrictions from competitive sport, particularly in children and adolescents with mild aortic dilation (z score < 3), which in turn may negatively affect physical and psychological well-being, leading to a sedentary life. However, several studies do not confirm an unfavorable effect of training, and particularly of regular exercise, on aortic enlargement or BAV dysfunction [41,42]. Although a propensity towards progressive dilation of the ascending aorta is also described in paediatric ages and in adolescence, regular exercise seems not to be an independent risk factor for progression [43].

### 6.3. Pre-Pregnancy Counselling in Adolescents with BAV

Uncomplicated BAV should not be considered a risk factor for future pregnancy and does not require specific advice during adolescence, apart from information on possible recurrence in the offspring. 

On the contrary, in young patients with stenotic and/or insufficient BAV, or with associated aortic dilation, potential pregnancy-related risks should be discussed in their early adolescence, during the phase of ‘transition’ of care [44]. Moreover, young females with moderate AS and/or dilation of the ascending aorta should be advised about the possible need for corrective surgery before pregnancy. Pregnancy should be avoided when the aorta diameter is >50 mm [45]. Problems during pregnancy related to mechanical or biological prostheses must be considered when surgery is planned.

Comprehensive advice about contraception should be provided for adolescents with BAV (as for all young patients with CHD). Particularly, the risk of IE associated with the use of intrauterine devices should be considered in patients with stenotic or insufficient BAV.

## 7. Therapy

### 7.1. Balloon Valvuloplasty

BAVs requiring treatment for severe stenosis at birth or in the first month of life represent a peculiar subgroup and are not discussed in this review. 

BAV in children and adolescents may be stenotic or incompetent; when required, surgery is more often the therapeutic choice. However, the optimal initial management for stenotic BAV is still controversial and the percutaneous approach is considered a valuable alternative. Balloon aortic valvuloplasty is a palliative strategy first described in 1983 by Lababidi [46]. It has the aim of reducing the transvalvular gradient in order to postpone the need for surgery [47]. 

Balloon aortic valvuloplasty can be performed by an “anterograde” approach via the femoral vein reaching the aortic valve through an atrial septal defect or, more usually, by a “retrograde” one, via the femoral artery or carotid or brachial artery [48]. The main complication of the procedure is represented by secondary AR; because the risk is increased when oversized balloons are used [49], the balloon catheter is chosen to obtain a balloon/annulus ratio less than 0.9. Potential complications also include access site injuries, stroke, rhythm disturbances, rupture of valves or myocardial perforation [50]. The procedure is considered effective when reduction of the transvalvular gradient is achieved without the development of more-than-mild AR or worsening of pre-existing AR [49].

The short- and long-term results of the procedure can be influenced by several factors. Careful selection of cases, with an accurate assessment of valve morphology, together with the level of the operator’s expertise, may improve the results [51]. Mixed valve disease, baseline gradient more than 60 mmHg and baseline AR greater than mild and multiple balloon inflation are some of the factors described as predictive of suboptimal result [49]. Furthermore, although not universally confirmed, an association between aortic valve morphology and the need of reintervention has been proposed: Maskatia et al. observed that BAVs were associated with higher freedom from reintervention than UAVs; they also described that true BAV developed post-procedural AR more frequently than the functionally bicuspid or unicuspid valves [52].

Compared to surgery, balloon aortic valvuloplasty is less invasive but may confer a higher risk of reintervention as a consequence of both restenosis and aortic insufficiency. Lower post-procedural gradient and milder post-procedural AR were associated with longer freedom from aortic valve replacement, and a favorable outcome was observed in cases of residual mean gradient less than 35 mmHg [49,53]. Patients with moderate/severe AR and residual gradients <35 mmHg experienced longer freedom from aortic valve replacement than patients with mild AR but a higher residual gradient [54]. At a mean follow-up of 10 years, survival free from aortic valve surgery can reach 70% for patients who underwent balloon valvuloplasty [55]. However, when comparing surgery to balloon valvuloplasty, the literature is still not in agreement. Although some authors have observed a longer freedom from reintervention after surgical aortic valvulotomy rather than after balloon aortic valvuloplasty [56,57], others have suggested a similar survival free from aortic valve replacement after both the procedures [58]. A high rate of successful balloon valvuloplasty has been confirmed in adolescents and young children with non-calcific BAV at a mean follow-up of 5.7 ± 1.3 years [59].

Balloon aortic valvuloplasty appears to be a safe and effective palliative strategy in selected patients with congenital AS, with satisfactory short- and long-term results and low periprocedural mortality.

### 7.2. Surgery

Surgery for BAV has evolved dramatically in the last two decades [60]. This improvement is due principally to the comprehension of aortic root function and the interactions of all its components being approached simultaneously during surgery to achieve the most durable results of bicuspid aortic valve repair and postpone aortic valve replacement nowadays [60,61]. Surgery for BAV also includes the management of ascending aorta aneurysms that are present in almost 50% of patients. The ascending aorta may be considered anatomically the “keystone” of aortic root architecture, playing an important role in its function [62]. 

In neonates and infants, AS is more frequent than AR. In recent years, balloon valvuloplasty has been widely adopted as primary line treatment for severe AS in for this age [63]. In all children that have not reached definitive somatic growth, the need to preserve annular development and to avoid valve replacement makes the management of aortic valve disease more challenging. In recent years, the advances in surgical techniques have led to comparable results between balloon valvuloplasty and surgery in terms of mortality and freedom from reintervention [63,64]. Several aortic valve repair techniques, especially by the use of pericardial patch reconstruction of the aortic cusps, have allowed management of more complex lesions. Most of these techniques have been inherited from adult surgery; however, their use in children may allow aortic native valve sparing, so postponing the Ross operation or aortic valve replacement. 

In paediatric patients, the Ross operation has shown excellent mid- to long-term results in terms of mortality and freedom from re-operations. Ivanov and colleagues reported an early mortality of 1.3% with only one death occurring in a neonate [65]. Freedom from reintervention was 90% at 10 years. As well as aortic valve repair, the Ross operation carries the advantage in this period of life of the autograft (and hence neo-aortic annulus) being able to grow with the patients.

On the other hand, the Ross operation transforms one-valve disease into two-valve disease, with the drawbacks related to pulmonary valve replacement by homograft or artificial conduit in the smallest children. Danial and colleagues compared patients undergoing to complex aortic valve repair with patients undergoing the Ross operation with propensity score matching. The authors highlighted some important advantages in the group managed by aortic valve repair; they reported a lower morbidity, probably due to the minor complexity of the repair procedure versus the Ross operation, and a lower overall incidence of IE [66]. Therefore, several authors suggest postponing the Ross operation beyond infancy, ideally into adulthood, and recommend aortic valve repair in growing children [64]. 

Regarding surgical techniques, BAV repair carries many advantages, especially avoiding lifelong anticoagulation and the drawbacks related to haemorrhagic or thrombo-embolic events [67,68]. The durability of bicuspid valve repair is determined by different anatomic and surgical variables: annulus, cusps, sinus of Valsalva and ascending aorta, use of patches or artificial materials. Since BAV is primary an anomaly of the aortic cusps, cusp repair is undoubtedly the primary step in BAV repair. Intraoperative assessment of cusp morphology must be correlated with echocardiographic findings. Cusp morphology, amount and tissue quality, orientation of the commissures and geometric and effective height must be carefully assessed both visually and by use of a specific calliper developed for this type of surgery. An effective height of 8–9 mm and a geometric height of >20 mm are considered parameters for durable results after BAV repair [69,70]. 

Because cusp prolapse is much more frequent than retraction, plication of the free cusp margin has been widely adopted to manage it. Central plication of the free margin is reproducible and has the purpose of equalizing the line of coaptation of the cusps, improving the effective and geometric height. Sometimes, in case of cusp calcification, complex cusp reconstruction may be required with the excision of the calcified portion and re-approximation of cusp segment by interrupted or continuous sutures. In this case, the amount of tissue plays an important role in avoiding the use of additional patches, which have been demonstrated to reduce the durability of repair due to major susceptibility of the patches to calcification [60,71,72]. After cusp manipulation, the root configuration must be approached. Commissural orientation plays an important role and a commissural angle between 160–180° is considered as the optimal configuration associated with better root fluid dynamics [60,69,70,71,72]. The cusp plications described above permit modification of the commissural angle. However, sometimes root reimplantation or root remodelling are required to re-establish an optimal commissural angle. Root reimplantation has the advantage of managing eventual annular dilatation at the same time; however, the advantage of this approach instead of cusp repair and concomitant annuloplasty is not yet demonstrated despite recent studies demonstrating similar results with root reimplantation in bicuspid and tricuspid valves [73,74].

The same recommendations have been proposed in cases of aortic annular dilatation. Annular dilation has been identified as an independent risk factor for progression of aortic regurgitation in BAV. The measure of 25–27 mm was identified as the cut-off for a mandatory aortic annuloplasty [60]. There are several surgical techniques to approach aortic annular dilatation. Early, sub-commissural plication sutures have been adopted to reduce aortic annulus. However, several studies have demonstrated poor results, leading to abandonment of this technique [60]. Currently, annuloplasty is performed by several techniques. Basal ring (i.e., a plane passing at the nadir of aortic cusps) annuloplasty may be performed by ePTFE sutures placed at the basal rings and tied around a Hegar dilator [75]. Alternatively, an external sub-coronary ring made from a Dacron or ePTFE conduit or an internal geometric pre-formed ring may be used for annular stabilization [76].

Ascending aorta dilatation must be taken into consideration during BAV surgery. Current guidelines recommend ascending aorta replacement in cases of a diameter > 45 mm if aortic valve surgery (repair or replacement) is planned [77]. Ascending aorta replacement, even if mildly dilated, has been associated with the durability of BAV repair [60,77].

By the use of this current tailored approach in BAV repair, surgical results have markedly improved in the recent years. Several groups have demonstrated a freedom from reoperations of 90% at 10 years [78]. Aortic valve replacement remains a valid alternative, especially when all the anatomic characteristics described for the feasibility of the repair are not present. 

## 8. Conclusions

Isolated BAV is a common finding in children and adolescents; although it is usually a benign condition, a significant (more than moderate) valve dysfunction is reported in a minority, requiring surgery or interventional procedures. The association with aortic dilation is common and progression is also reported in paediatric ages. Careful follow-up is therefore needed. Although a genetic basis is largely recognized, BAV is a multifactorial and polygenic disorder, and genetic testing remains inconclusive in the majority of cases.

A comprehensive 2D TTE study is usually adequate in children and adolescents to completely assess BAV morphology and function and to exclude associated cardiovascular lesions. However, CMR is increasingly used also in paediatric patients, especially when BAV is associated with aortic dilation, in order to evaluate aortic diameters and flow abnormalities. 

The rate of controls during follow-up should be tailored on the basis of the presence and degree of valve dysfunction and aortic dilation. Paediatric patients with normally functioning, uncomplicated BAV can undergo clinical and echocardiographic controls every 2 years.

Children and adolescents with uncomplicated BAV require no limitations of sport activity.

Infective complications are exceedingly rare, and antibiotic prophylaxis is not currently indicated, mainly in normally functioning bicuspid valves; however, young patients should be advised against tattoo and piercing, and careful oral hygiene is strongly recommended.

As the majority of complications occur in adults, a transition program of care should be established to provide adequate maintenance of care, especially in young patients with complicated BAV.

## Figures and Tables

**Figure 1 diagnostics-12-01751-f001:**
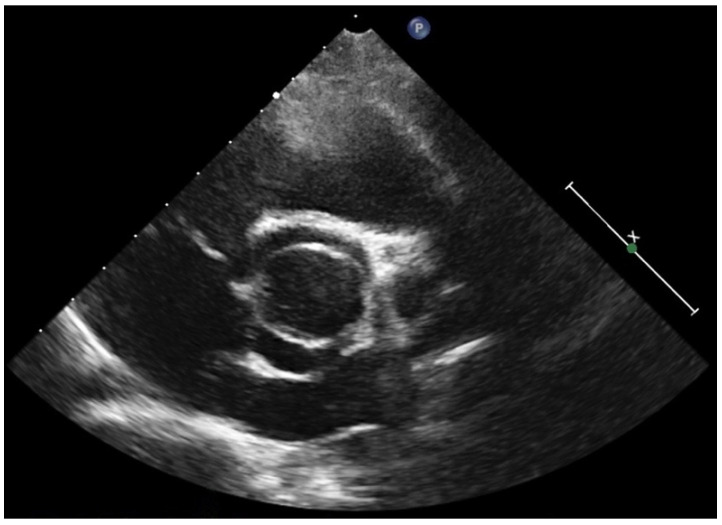
2D Echocardiography parasternal short axis view (systolic frame) at the level of aortic valve shows a bicuspid aortic valve in an 8-year-old patient.

**Figure 2 diagnostics-12-01751-f002:**
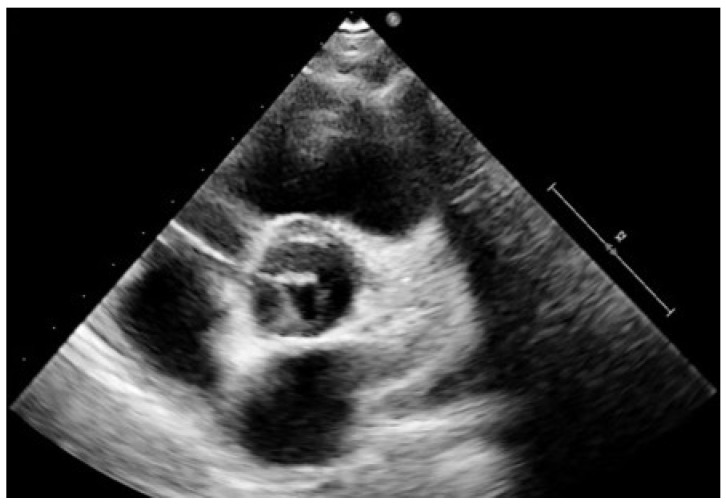
2D Echocardiography parasternal short axis view (systolic frame) at the level of aortic valve shows an unicuspid aortic valve in a 12-year-old patient.

**Figure 3 diagnostics-12-01751-f003:**
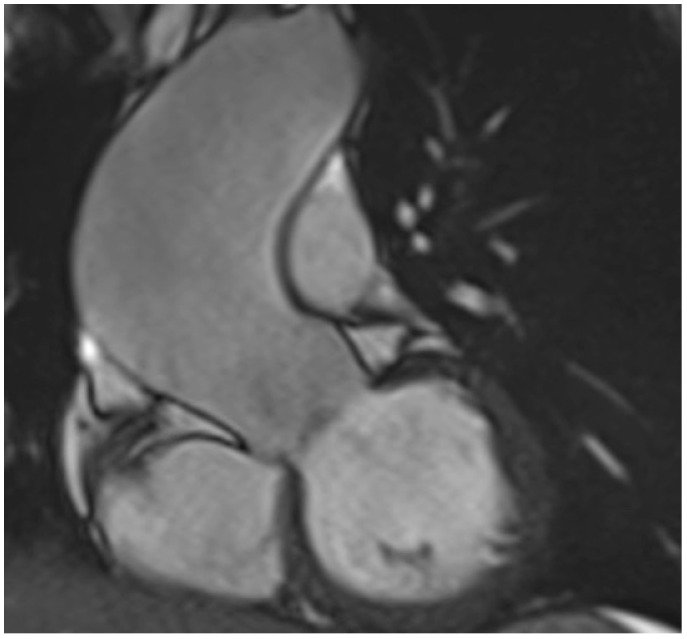
Cardiac magnetic resonance imaging of the left ventricular outflow tract shows a dilation of the ascending aorta in a 16-year-old patient with bicuspid aortic valve.

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
