# Peer review of "Bicuspid Aortic Valve in Children and Adolescents: A Comprehensive Review"

_diagnostics, 2022, doi:10.3390/diagnostics12071751_

Round 1

Reviewer 1 Report

Dr.Spaziani et al.'s review 'Bicuspid aortic valve in children and adolescent: a comprehensive review' is well written and constructed of very understandable descriptions.

However, an essential issue of the BAV patient's pregnancy is not mentioned. It is quite a concerning problem for female patients. If there are not sufficient references or clinical experience to summarize, please describe such a situation. It is necessary to talk with young female patients with BAV, including aortopathy. 

Line 122: What is 'VAB'? No explanation for its first appearance. Is it confusing with 'BAV,' different or the same?

Reviewer 2 Report

I read with interest the article by Spaziani et al. entitled «Bicuspid aortic valve in children and adolescent: a comprehensive review»

Though well written and timely studied, I believe that the manuscript should be improved.

The major point of discussion in this article is that, as the authors acknowledged, the « untouched » (no neonatal dilatation) BAV is rarely symptomatic in children and young adolescent so that most of the paragraphs discussing endovascular dilatation and BAV repair applies more to young adults.

For those young adult patients, the paragraph on repair is well written but as the title refers to children and adolescent, I believe that the authors should have spend more time on the specific bibliography of aortic valve surgery in the 1-14 year range, with specific emphasis on BAV.

The litterature will be scarse, but the point should be : when the aortic annulus has not yet reached a sufficient value for a lifelong stabilization (external, internal, reimplantation, whatever..) : what is the best approach ?

Also, not sufficient emphasis is made on the functionnal evaluation of the LV : MRI for the indexed systolic volume for decision making, not did the authors discuss the Strain evaluation by Echo which is also an important tool when balancing between « Await to allow growth, and go for surgery »

Clearly, indication for treatment should be refined if the « litterature review « is the goal of the manuscrpit.

Besides :

-Prevalence of the association Ao Coarctation & BAV is somewhat exagerated, and most studies agree for a prevalence of around 50%.

-Line 77, Line 121 and Line 174, the abbreviation VAB is written, I guess it stands for BAV ??

-Line 106 bebore instead of before…..

I enjoy reviewing this manuscript which has room for improvement...

Round 2

Reviewer 1 Report

Our comments are considered and responded to sufficiently.

Reviewer 2 Report

Dear Authors,

The manuscript has been improved since its initial version.

I would still recommend minor changes (syntax) as well as some additional data/information

Syntax-editing-spelling

Line 314                       “are” instead of “were”

Line 317                       “are” instead of “were”

Line 336                       “Techniques” instead of “technics”

Line 373                       “Early on” instead of “In the last year”

Sequence of paragraph

Lines 365-368  which deal with the ascending aorta in the context of Root/valve surgery : this paragraph should be placed AFTER the one which deals with management of the annulus.

Additional data/information

Line 334 There should be more discussion and more references on the different techniques of valve repair in children below 12-14 (below puberty) who present with

a)      Post-balloon PTA severe AR, or moderate and impaired GLS

b)      Dysplastic stenotic aortic valves

Several authors have recently emphasized that before the aortic annulus has reached a “reasonable” size (grossly 22 for predicted small height, 24 for others), most efforts to repair a valve should be done, the Ross remaining a bail out in cases of failure as well as in unrepairable valves.

(references :

-Aortic valve repair in pediatrics-time to swing the pendulum back?

d'Udekem Y. Ann Cardiothorac Surg. 2019 May;8(3):396-398.

-Ross procedure or complex aortic valve repair using pericardium in children: A real dilemma

Pichoy Danial,Asma Neily,Margaux Pontailler,...Mary Osborne-Pellegrin,Pascal Vouhe,Olivier Raisky. JTCVS March 2022

And others..
